# Symbolic or Numerical? Understanding Physics Problem Solving in Reasoning LLMs

## Abstract

Navigating the complexities of physics reasoning has long been a difficult task for Large Language Models (LLMs), requiring a synthesis of profound conceptual understanding and adept problem-solving techniques. In this study, we investigate the application of advanced instruction-tuned reasoning models, such as Deepseek-R1, to address a diverse spectrum of physics problems curated from the challenging SciBench benchmark. Our comprehensive experimental evaluation reveals the remarkable capabilities of reasoning models. Not only do they achieve state-of-the-art accuracy in answering intricate physics questions, but they also generate distinctive reasoning patterns that emphasize on symbolic derivation. Furthermore, our findings indicate that even for these highly sophisticated reasoning models, the strategic incorporation of few-shot prompting can still yield measurable improvements in overall accuracy, highlighting the potential for continued performance gains.

## 1 Introduction

Recent advances in large language models (LLMs), particularly models such as GPT-O1 and DEEPSEEK-R1, have substantially improved the capabilities of numerous complex reasoning tasks OpenAI (2023); Chung et al. (2022). Historically, researchers have used a wide range of specialized methods and sophisticated prompt engineering techniques, including chain-of-thought prompting Wei et al. (2022), structured few shot prompting Brown et al. (2020), and retrieval-augmented generation Lewis et al. (2020) to improve LLM performance in challenging domains such as physics.

Despite their success, these traditional approaches typically incur significant effort in the design of domain-specific prompts and the maintenance of auxiliary systems. Moreover, the performance of these approaches can vary widely depending on the effectiveness of prompt construction and the availability of external computational tools Madaan et al. (2023); Huang et al. (2023). Consequently, there is an ongoing demand to explore simpler yet equally effective strategies to leverage the inherent reasoning capabilities of modern LLMs, particularly as these models continue to grow in size and sophistication Kaplan et al. (2020).

The advent of advanced reasoning-focused models has raised important questions about the necessity and efficiency of these complex engineering efforts. These recent models are specifically optimized through extensive instruction tuning and reinforcement learning from human feedback Ouyang et al. (2022); Taori et al. (2023), enhancing their native ability to reason logically and coherently without relying heavily on external assistance. In this work, we empirically investigate whether contemporary instruction-tuned reasoning models can independently achieve high performance on physics reasoning tasks without extensive prompt engineering or external augmentation and what reasoning mechanisms underlie their behavioral divergence from standard chat models. Additionally, we seek to determine whether carefully designed few-shot prompt engineering continues to provide measurable benefits for advanced LLMs in the physics domain.

We evaluate the DEEPSEEK-R1 and its distilled models across three representative physics datasets from the SciBench benchmark Chen et al. (2023), covering fundamental topics such as classical dynamics, thermodynamics, and fundamental physics and comprising diverse and challenging problems.

Our findings demonstrate that reasoning-focused LLMs alone attain satisfactory results, achieving competitive accuracy on challenging physics problems. Furthermore, we show that targeted few-shot prompts can still enhance the performance of advanced models, providing valuable improvements in accuracy and interpretability. Moreover, our study reveals distinctive reasoning patterns by analyzing the chain-of-thought (CoT) outputs generated by different types of models. We observe that reasoning-specialized models prefer symbolic derivation—algebraically manipulating equations before numeric substitution—to solve physics calculation problems in most cases, in contrast to chat-oriented models that rely on procedural, step-by-step numerical substitution. This divergence highlights symbolic reasoning as a distinguishing factor contributing to the accuracy and robustness of reasoning-specialized models in multi-step scientific problem-solving tasks.

## 2 Related Work

### 2.1 Physics Problem Solving with LLMs

Early efforts to apply large language models (LLMs) to physics reasoning treated textbook-style questions as pure text-completion tasks. For example, Gao et al. Gao et al. (2022) evaluated GPT-3 on introductory mechanics and electromagnetism problems, finding limited success with zero-shot prompting, especially on multi-step derivations. To improve performance, Wei et al. Wei et al. (2022) introduced a prompt *chain-of-thought*, demonstrating substantial gains in math and logic benchmarks; subsequent work by Kojima et al. Kojima et al. (2022) extended these benefits to physics questions.

More recent approaches combine LLMs with external tools. Program-aided language models (Liu et al. Liu et al. (2023)) integrate symbolic solvers for arithmetic and algebraic steps, while tool-augmented frameworks (Huang et al. Huang et al. (2023)) call unit conversion libraries and equation solvers via APIs. Self-verification techniques (Nye et al. Nye et al. (2024)) further enhance reliability by having the model re-check its solution steps against physical laws. These methods, however, require additional infrastructure or fine-tuning. In contrast, our work examines the power of *in-context* prompt design alone—without external tools or parameter updates—to boost pure physics reasoning in state-of-the-art instruction-tuned models.

### 2.2 Prompt Engineering and Advanced Language Models

The paradigm of *few-shot prompting* was popularized by Brown et al. Brown et al. (2020), who showed that adding exemplars in the prompt can dramatically improve LLM performance. Based on this, the decomposition prompts (Madaan et al. Madaan et al. (2023)) explicitly break problems into sub-questions within the context. As LLMs have been refined through instruction tuning (Chung et al. Chung et al. (2022)), reinforcement learning from human feedback (Ouyang et al. Ouyang et al. (2022)) and specialized reasoning curricula (Smith et al. Smith et al. (2023)), the marginal gains from complex prompts have come under scrutiny.

Zheng et al. Zheng et al. (2024) evaluated prompt variants in GPT-4 code generation, finding that simple zero-shot prompts often matched or outperformed elaborate few-shot templates. Li et al. Li et al. (2024) similarly observed that instruction-tuned models can produce high-quality reasoning chains without exemplars on logic puzzles. However, these studies focus on general coding or reasoning benchmarks rather than domain-specific tasks. Our paper fills this gap by systematically studying few shot physics prompts in advanced reasoning models, demonstrating that carefully chosen exemplars continue to yield significant accuracy improvements in physics problem solving.

## 3 Experiment

### 3.1 Overview

Our experimental workflow, as Figure 1 illustrates, systematically assesses the problem-solving capabilities of reasoning-tuned LLMs on physics questions. We begin by selecting a representative set of problems from the SciBench Chen et al. (2023) benchmark, encompassing mechanics, thermo-dynamics, and electromagnetism, and formatting each into a standardized prompt. For every problem, we generate both a Zero-Shot CoT prompt and a Few-Shot CoT prompt. We then run these prompts

through our reasoning models and baseline chat models in parallel to compare their performance in terms of accuracy and error categories. During inference, we record the complete Chain-of-Thought outputs for both reasoning and chat models to evaluate not only the final answer but also the quality of their intermediate reasoning steps.

## 3.2 DATASETS

We conduct experiments using three representative datasets from the SciBench benchmark. After filtering out problems that require detailed solutions and visual components, we focus exclusively on textbook-style questions. The resulting datasets are summarized in Table 1.

| Dataset | Field | # P | # S |
|---------|-------|-----|-----|
| fund | fundamental physics | 71 | 10 |
| thermo | thermodynamics | 66 | 17 |
| class | classical dynamics | 56 | 7 |

Table 1: Dataset statistics after filtering out problems with visuals. #S denotes the number of available detailed solution per subset.

**Dataset Selection.** We selected the dataset from SciBench due to its challenging nature: solving these problems requires not only scientific literacy but also strong reasoning skills, including complex calculations and step-by-step logical deduction. This effectively distinguishes model capabilities. Moreover, the dataset spans diverse fields and ranges from three different physics fields: electronics, thermodynamics, and classical dynamics.

**Dataset Filtering.** Since some baseline chat models lack multimodal capabilities, we exclude problems containing visual elements and focus solely on textual problems. Additionally, we filter out problems with detailed solutions to ensure they can be used as few-shot prompts.

## 3.3 MODELS

**Selected Model.** In the experiment, we select Deepseek-R1 and its distilled models R1-distill-LLaMA-70B and R1-distill-Qwen-32B. They are highly efficient open-weight models designed to balance strong reasoning capabilities with reduced computational costs. DeepSeek-R1 demonstrates robust performance in complex reasoning tasks, while its distilled versions maintain competitive ability, leveraging knowledge transfer from larger teacher models (LLaMA-70B and Qwen-32B) to achieve cost efficiency. The distillation process optimizes inference speed and memory usage, allowing R1-distill variants to deliver cost-effective alternatives while retaining core logical and analytical strengths.

**Baselines.** We compare the results of our models against baseline performances reported in the SciBench benchmark Chen et al. (2023). SciBench evaluates the reasoning capabilities of a wide range of general-purpose large language models across various physics domains using a unified framework. The benchmark includes standard instruction-tuned models such as LLAMA-2 (7B and 70B), MISTRAL-7B, CLAUDE2, GPT-3.5-TURBO, GPT-4, and GPT-4-TURBO, assessed under both zero-shot and few-shot Chain-of-Thought (CoT) prompting settings. In doing so, we aim to make a comprehensive comparison of the reasoning capability between reasoning models and chat-based models.

**Parameters Setup.** In our implementation, parameters are configured to ensure stable and reproducible model inference. We set the temperature to a near-zero value (1e-30) to eliminate sampling variability, thereby enforcing deterministic behavior and ensuring consistency across repeated runs. The number of returned completions is set to one (n=1), as our evaluation focuses on top-1 performance. To enhance robustness, the retry mechanism is configured with a high tolerance for failure: the patience parameter is set to $10^9$, allowing the system to persist through transient API issues without manual intervention.

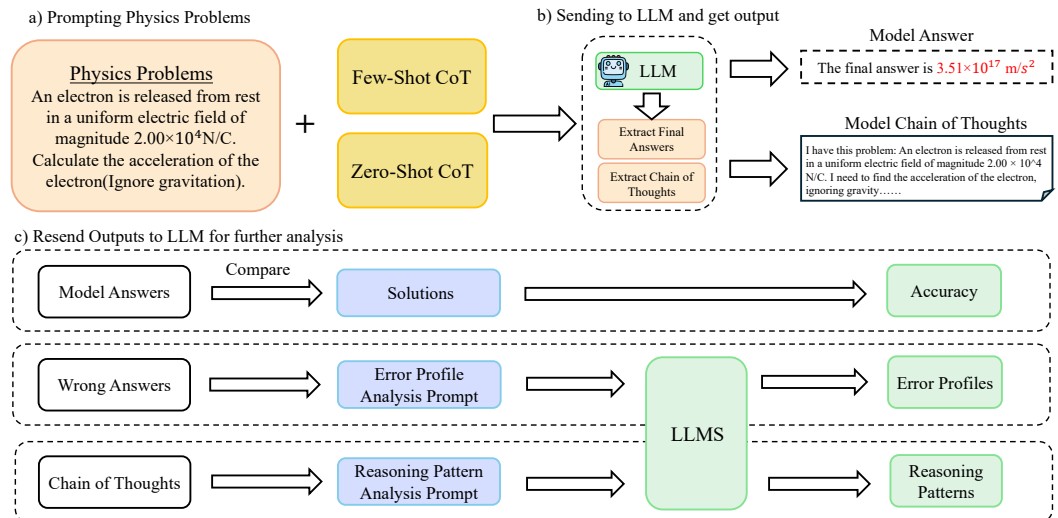

Figure 1: Overview of the experimental pipeline. A diverse set of physics problems is sampled from three domains: Fund, Thermo, and Class. Each problem is fed into the model under two prompting conditions: (1) Zero-shot Chain-of-Thought (CoT) prompting, and (2) Few-shot CoT. The model's output solutions and CoT traces are evaluated along two axes by sending back to LLMs: error categorization (analyzing incorrect reasoning types) and reasoning pattern analysis (identifying characteristic cognitive strategies).

## 3.4 PROMPTING CONDITIONS

We test model performance under two distinct prompting strategies:

- **Zero-Shot CoT.** The model is prompted to "think step by step" before answering, but receives no prior examples. The goal is to test whether the instruction-tuned reasoning capabilities of DEEPSEEK-R1 are sufficient to generate coherent multi-step solutions without external exemplars.
- **Few-Shot CoT.** The prompt is prepended with a few example problems, drawn from existing dataset instances that include detailed solutions. For controlled experimentation, we select only the top three exemplars from each subset.

These strategies allow us to analyze not only the final accuracy, but also the structure and correctness of the intermediate reasoning steps, which is critical to understanding whether errors stem from flawed logic or mere computational missteps. The zero-shot approach highlights the model's intrinsic reasoning capabilities, while the few-shot setting measures its ability to adapt to demonstrated solution patterns. This dual evaluation provides deeper insights into the model's problem-solving robustness beyond surface-level metrics.

## 3.5 EVALUATION METHOD

**Accuracy.** The evaluation of the accuracy of the solution is carried out by comparing each numerical response generated by the model with the reference value within a fixed relative tolerance of 5%. A response is considered to be correct if its parsed value falls within the specified tolerance of the ground-truth answer.

**Error Categories.** To better understand the shortcomings in incorrect solutions, we analyzed error types using the *SciBench* error automatic categorization framework,which employs an LLM to verify incorrect solutions and classify the error type of each one. This allows us to identify key reasoning gaps and assess the strengths of different models

**Chain-of-thought Output.** Chain-of-thought outputs from LLMs are collected by embedding three exemplar problem–solution pairs. The model's subsequent output—which interleaves reasoning

| Model | Zero-Shot + CoT Wei et al. (2022) | | | | Few-Shot + CoT Wei et al. (2022) | | | |
|---|---|---|---|---|---|---|---|---|
| | Fund | Thermo | Class | Avg | Fund | Thermo | Class | Avg |
| LLaMA-2-7B | 0.00% | 0.00% | 0.67% | 0.22% | 1.87% | 5.48% | 3.60% | 3.65% |
| LLaMA-2-70B | 0.93% | 0.00% | 1.89% | 0.94% | 13.10% | 12.33% | 8.40% | 11.28% |
| Mistral-7B | 6.54% | 0.00% | 4.63% | 3.72% | 6.54% | 2.13% | 6.09% | 4.92% |
| Claude2 | 20.56% | 3.08% | 10.99% | 11.54% | 15.89% | 6.12% | 15.26% | 12.42% |
| GPT-3.5-Turbo | 6.54% | 10.20% | 12.19% | 9.64% | 8.41% | 6.12% | 11.99% | 8.84% |
| GPT-4 | 28.04% | 20.41% | 25.37% | 24.61% | 41.12% | 16.33% | 25.36% | 27.60% |
| GPT-4-Turbo | 60.75% | 28.57% | 42.37% | 43.90% | 59.81% | 18.37% | 39.45% | 39.21% |
| Deepseek-V3 | 63.40% | 50.00% | 65.20% | 59.53% | 53.50% | 32.10% | 25.80% | 37.13% |
| R1-distill-LLaMA-70B | 64.80% | 55.40% | **68.20%** | 62.80% | 62.00% | 50.00% | 66.70% | 59.57% |
| R1-distill-Qwen-32B | 76.10% | 74.50% | 66.70% | 72.43% | 74.60% | 65.20% | 51.80% | 63.87% |
| Deepseek-R1 | **88.70%** | **76.50%** | 62.50% | **75.90%** | **93.00%** | **66.10%** | **84.80%** | **81.30%** |

Table 2: Physics accuracy (%) on the *fund*, *thermo*, and *class* domains under Zero-Shot and Few-Shot CoT prompting for models ranging from LLaMA-2-7B through GPT-4-Turbo. **Bold** indicates the best result per column; underline, the second-best. Model performance data from LLaMA-2-7B through GPT-4-Turbo is drawn from the SciBench benchmark. Chen et al. (2023).

steps with the final boxed answer—is recorded in full for each instance. Downstream analysis then proceeds by closely examining representative correct and incorrect reasoning chains to identify systematic inferential faults and reasoning patterns. This process involves human reviewers analyzing the CoT outputs to discern distinct reasoning patterns, and subsequently designing prompts that guide LLMs to analyze solutions for classification of reasoning patterns.

# 4 RESULTS

## 4.1 PERFORMANCE ACROSS DATASETS

**Comparison to Baseline Models.** Compared to general-purpose models like GPT-4, Claude2, and GPT-3.5-Turbo, the R1-series models (Deepseek-R1 and its distilled versions) show a clear advantage in physics-related tasks. For instance, in zero-shot mode, Deepseek-R1's average accuracy (75.9%) was nearly double that of GPT-4-Turbo (43.9%), with particularly large gaps in Fund (88.7% vs. 60.75%). The distilled models maintained competitive performance while potentially offering better computational efficiency, suggesting that model distillation can retain high accuracy while reducing resource demands.

**Impact of Few-Shot Prompting.** Our experiments reveal that few-shot prompting continues to offer tangible benefits, even for models that already exhibit strong zero-shot reasoning capabilities. For instance, DEEPSEEK-R1's performance improves further to 81.3% with the inclusion of few-shot CoT exemplars, demonstrating that carefully constructed demonstrations enhance reasoning quality and stability. This trend is particularly evident in the classical mechanics domain, where the few-shot accuracy rises from 62.5% to 84.8%.

## 4.2 PERFORMANCE IN DIFFERENT DOMAINS

**Thermodynamics.** Thermodynamics emerged as the most challenging domain, presenting unique difficulties even for top-performing models. Deepseek-R1's 76.5% zero-shot accuracy in thermo, while respectable, represents a significant drop from its fundamental physics performance. Notably, few-shot prompting provided minimal improvements in this domain, suggesting that thermodynamics' abstract, multi-step conceptual problems resist straightforward example-based learning.

**Fundamental Physics.** In fundamental physics, models achieved their strongest results, with Deepseek-R1 reaching 88.7% (zero-shot) and 93.0% (few-shot) accuracy. This superior performance aligns well with large language models' inherent strengths in pattern recognition and mathematical manipulation.

**Classical dynamics.** Classical dynamics showed the most dramatic response to few-shot learning techniques, offering encouraging insights about model adaptability. Deepseek-R1's performance in this domain jumped from 62.5% (zero-shot) to 84.8% (few-shot), indicating that classical mechanics' concrete, iterative problems are particularly amenable to contextual learning.

### 4.3 MODEL SIZE VS PERFORMANCE

In R1-distill models, The smaller R1-distill-Qwen-32B consistently outperforms its larger counterpart R1-distill-LLaMA-70B across most physics benchmarks, achieving superior scores in fundamental physics (76.1% vs. 64.8% zero-shot) and thermodynamics (74.5% vs. 55.4%). This result demonstrates that the Qwen architecture's superior symbolic processing capabilities more than compensate for its reduced parameter count. The performance advantage is particularly notable given the 32B model's significantly lower computational requirements.

The results also reveal that mid-sized distilled models rival much larger generalist models (e.g., GPT-4), demonstrating that task-specific optimization outweighs pure scaling. Full-sized models like Deepseek-R1 still dominate in few-shot learning, suggesting that parameter count remains critical for in-context learning flexibility. Notably, R1-distill-Qwen-32B cuts match scores or even outperforms larger models like GPT-4 in throughput while preserving high-quality chain-of-thought reasoning. The performance advantage is particularly notable given the 32B model's significantly lower computational requirements. Therefore, distilled models strike an optimal balance between performance and resource utilization.

### 4.4 ERROR REDUCTION CATEGORIES

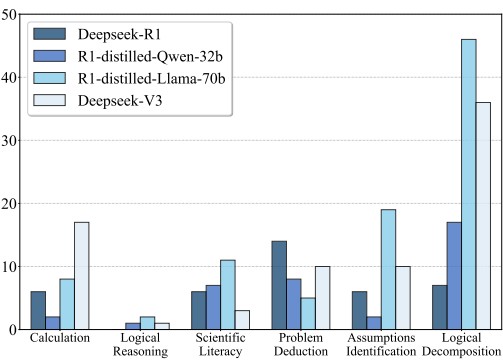
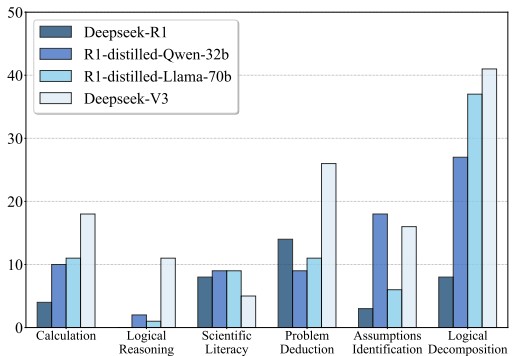

(a) Error distribution across SciBench categories (zero-shot CoT).

(b) Error distribution across SciBench categories (Few-Shot CoT).

Figure 2: Comparison of error distributions across different prompting methods, excluding near-zero categories

As Figures 2a and 2b show, We also analyze the performance improvement of Deepseek-R1 over Deepseek-V3, as well as few-shot over zero-shot prompting under R1, using error categories based on the essential scientific problem-solving skills defined in SciBench. (Note: SciBench includes 10 error categories, but we exclude the zero and near-zero categories to focus on the major ones). The following categories have shown clear error reductions.

**Logical Decomposition.** Deepseek-R1 maintains exceptional performance with minimal degradation from 7 to 8 errors (14.3% increase), demonstrating robust logical decomposition capabilities that are largely independent of prompting strategy. R1-distilled-Llama-70b shows moderate improvement from 46 to 37 errors, representing a 19.6% improvement with few-shot examples. Deepseek-V3 experiences significant regression from 36 to 41 errors (13.9% increase), suggesting few-shot examples

**Problem**

A food shipper pushes a wood crate of cabbage heads (total mass $m = 14$ kg) across a concrete floor with a constant horizontal force $\vec{F}$ of magnitude 40 N. In a straight-line displacement of magnitude $d = 0.50$ m, the speed of the crate decreases from $v_0 = 0.60$ m/s to $v = 0.20$ m/s. Find the increase $\Delta E_{\text{th}}$ in the thermal energy of the crate and floor.

**Step-by-Step Numeric Substitution**

**Step 1: Compute Work Done by the Push**
The work done by the applied force is:
$W = F \cdot d = (40 \text{ N}) \cdot 0.5 \text{ m} = 20 \text{ J}$

**Step 2: Compute $\Delta K$**
$\Delta K = \frac{1}{2} m (v^2 - v_0^2)$
Substituting the given values:
$\Delta K = \frac{1}{2} \times 14 \text{ kg} \times ((0.20 \text{ m/s})^2 - (0.60 \text{ m/s})^2) = -2.2 \text{ J}$

**Step 3: Apply the Theorem**
The work-energy theorem states:
$W = \Delta K + \Delta E_{\text{th}}$
Solving for the change in thermal energy:
$\Delta E_{\text{th}} = W - \Delta K = 20 - (-2.2) = \boxed{22 \text{ J}}$

**Symbolic Derivation**

**Step 1: General Energy Relation**
The work-energy theorem, including thermal dissipation, states:
$\Delta E_{\text{th}} = W - \Delta K$
where:
$W = F \cdot d$
$\Delta K = \frac{1}{2} m (v^2 - v_0^2)$

**Step 2: Symbolic Substitution**
Substitute the expressions:
$\Delta E_{\text{th}} = Fd - \frac{1}{2} m (v^2 - v_0^2)$

**Step 3: Numerical Calculation**
Plug in the given values:
$\Delta E_{\text{th}} = (40)(0.50) - \frac{1}{2}(14)(0.20^2 - 0.60^2) = 20 - 7(-0.32) = \boxed{22 \text{ J}}$

Figure 3: Comparison of two solution strategies for finding the increase in thermal energy when a 14 kg crate is pushed 0.5 m by a 40 N force. **Left:** A step-by-step numeric approach, in which the work done by the push is calculated first, then the change in kinetic energy is determined, and finally the thermal energy increase is obtained by combining those results. **Right:** A symbolic approach, where a general expression for the thermal energy increase is derived in terms of work and kinetic energy change before the numerical values are inserted.

may introduce confusion for complex structural reasoning. R1-distilled-Qwen-32b shows the most dramatic decline from 17 to 27 errors (58.8% increase).

**Calculation Skills.** Few-shot prompting delivers mixed results across models. Deepseek-R1 achieves the best improvement, reducing errors from 6 to 4 (33.3% reduction), demonstrating enhanced arithmetic precision with worked examples. R1-distilled-Llama-70b shows modest improvement from 8 to 11 errors, while both Deepseek-V3 (17 to 18 errors) and R1-distilled-Qwen-32b (2 to 10 errors) exhibit performance degradation.

**Assumption Identification.** This category reveals the most pronounced few-shot benefits. Deepseek-R1 achieves a 50% error reduction from 6 to 3, demonstrating that exemplar-based prompting significantly enhances premise identification. R1-distilled-Llama-70b shows substantial improvement from 19 to 6 errors (68.4% reduction). However, both Deepseek-V3 (10 to 16 errors, 60% increase) and R1-distilled-Qwen-32b (2 to 18 errors, 800% increase) show concerning performance degradation, suggesting that few-shot examples may overwhelm these models' assumption-detection mechanisms.

## 4.5 REASONING PATTERN

To further analyze the behavioral differences between models, we examine the chain-of-thought outputs of correct answers from selected models and identify two predominant reasoning patterns used when solving physics problems across the three datasets(See Figure 3) :

**Step-by-Step Numeric Substitution.** This approach represents a direct computational approach in which solvers immediately replace variables with given numerical values. This method progresses linearly through arithmetic operations at each stage, moving efficiently from known quantities to final answers.

**Symbolic Derivation.** This approach embodies a more theoretical approach that maintains variables in their symbolic form throughout initial problem-solving stages. Solvers using this method to firstly establish complete mathematical relationships between quantities, then substituting numerical values in the final computation steps.

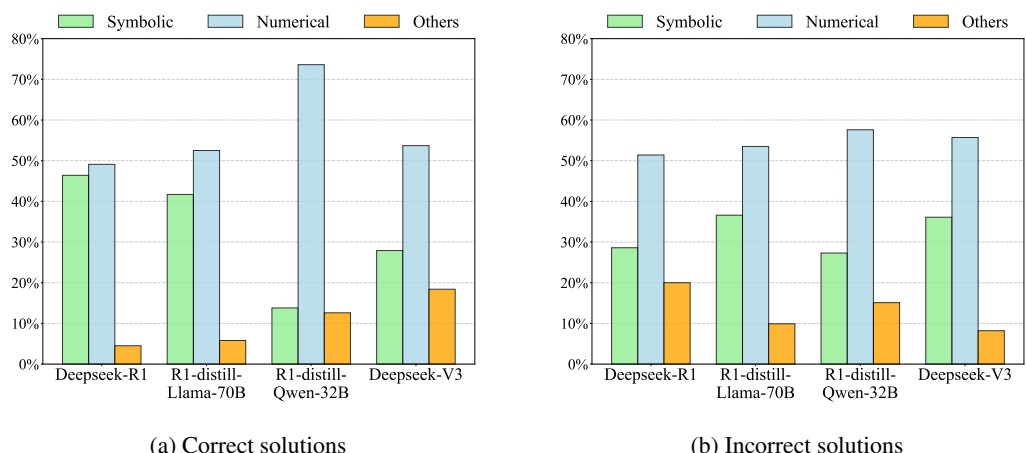

(a) Correct solutions            (b) Incorrect solutions

Figure 4: Distribution of reasoning patterns in (a) correct and (b) incorrect solutions across different Deepseek model variants.

## 5 CONCLUSION

This study investigates the capabilities of advanced reasoning-focused large language models (LLMs) in solving complex physics problems, with a particular focus on the instruction-tuned model DEEPSEEK-R1 and its distilled variants. Leveraging the SciBench benchmark, we systematically evaluate both zero-shot and few-shot Chain-of-Thought (CoT) prompting strategies.

Our results demonstrate that reasoning models consistently outperform general-purpose chat-based models across all datasets, even in zero-shot settings. Notably, DEEPSEEK-R1 achieves substantial improvements in both accuracy and interpretability, generating step-by-step solutions that reflect a deep conceptual understanding and precise symbolic manipulation. While few-shot prompting further enhances performance, its impact is less critical for such high-performing reasoning models. This finding suggests that although prompting strategies can still improve reasoning models, they already achieve satisfactory accuracy without external methods.Furthermore, we identify a clear dichotomy in reasoning patterns between specialized and chat-oriented models: reasoning-specialized models often employ symbolic derivation—algebraically manipulating equations prior to numeric substitution—particularly in correct solutions, while chat-oriented models, exemplified by Deepseek-V3, rely heavily on step-by-step numeric substitution, reflecting a more procedural and less abstract approach. This distinction provides critical insight into performance gaps observed in multi-step problems that demand abstract manipulation and structured reasoning. Collectively, these findings underscore the significance of symbolic reasoning as a key driver of robust performance, emphasizing the transformative potential of instruction-tuned reasoning models for physics education and complex scientific problem-solving tasks.

## 6 LIMITATIONS

Despite these promising findings, several limitations merit discussion. First, reasoning models such as DEEPSEEK-R1 incur substantial computational costs due to the verbose nature of CoT outputs. Compared to chat-oriented models, their step-by-step reasoning processes often result in significantly higher token counts—sometimes exceeding 10,000 tokens for a single problem(See Table 3). This increases inference latency and

Table 3: Average output tokens per model on the same question set.

| Model | Avg. Output Tokens |
|---|---|
| Deepseek–R1 | 14,698 |
| R1–distill (LLaMA–70B) | 7,688 |
| R1–distill (Qwen–32B) | 8,355 |
| Deepseek–V3 | 4,035 |

places a heavy burden on both memory and processing resources, potentially limiting scalability in real-world deployments or low-resource environments Kaplan et al. (2020); Zhang et al. (2023). Also, our analysis is limited to unimodal, text-only problems and does not account for questions requiring diagrammatic interpretation, spatial reasoning, or numerical simulation. Extending these models to multimodal inputs remains a future direction. Alayrac et al. (2022); Driess et al. (2023).

## REPRODUCIBILITY STATEMENT

We have taken several measures to ensure the reproducibility of our results. All datasets used in our experiments are drawn from the publicly available SciBench benchmark, and we clearly describe our filtering criteria and dataset statistics in Section 1. The model configurations, inference parameters, and prompting strategies (zero-shot and few-shot CoT) are specified in Section 3 and Appendix A, including temperature settings, retry mechanisms, and exemplar selection. Detailed error categorization procedures and reasoning pattern analyses are described in Section 4.4 and Appendix B, with examples provided to illustrate our methodology. To further support reproducibility, we include prompt templates, evaluation scripts, and implementation details in the supplementary materials. Together, these resources allow independent researchers to replicate our experimental pipeline and validate the reported findings.

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

## A EXPERIMENT DETAILS

### A.1 MODEL INVOCATION AND ROBUSTNESS

We wrap the OpenAI chat API call in a `Caller` class that: Uses a deterministic temperature ($T = 10^{-30}$). Retries up to a large patience count, with optional sleep between retries. Checks for non-empty responses before returning. Logs API errors to `stderr` and continues retrying.

### A.2 OUTPUT PARSING AND EVALUATION

The raw model output is post-processed to extract the numeric answer: Strip LaTeX boxing and other text. Normalize "not"-style units via `remove_not`/`cal_not`. Compare to ground-truth using `math.isclose` with 5% relative tolerance. Per-problem correctness is logged, and final accuracy is reported over the entire dataset.

### A.3 API USAGE

We instantiate the OpenAI client with the user's API key and OpenRouter base URL. We wrap each call in a retry loop with exponential backoff—initial sleep of 2 s doubling each retry—up to a maximum number of attempts. Before accepting a response, we validate that `response.choices` exists and contains non-empty `message.content`. Errors (network, rate limits, empty responses) are caught, logged, and trigger the backoff logic. This pattern ensures robust, deterministic interaction with OpenRouter while preserving per-problem logging and progress reporting.

### A.4 PROMPT CONSTRUCTION

To maintain the completeness and consistency of the experiment, the prompt construction follows the same format as the experimental setup used in SciBench Chen et al. (2023). For few-shot CoT evaluation, the prompt begins with a system message that defines the assistant's role (e.g., a helpful and accurate physics tutor), followed by several solved example problems. Each example includes a user query presenting the problem statement and an assistant response that provides both the step-by-step reasoning and the final boxed answer with units. The test problem is appended afterward without a solution. For zero-shot and zero-shot CoT settings, no examples are provided. Instead, the prompt contains only the system message and a single user query for the test problem. In the zero-shot CoT setting, we apply a two-stage prompting strategy: the first prompt elicits intermediate reasoning ("Let's think step by step"), and the second prompt feeds back this reasoning to request a final answer. All prompts include explicit unit information by appending "The unit of the answer is <unit>" to the problem text to reduce ambiguity and encourage unit-aware predictions.

### A.5 HUMAN EVALUATION

For reasoning pattern analysis, we involve human in the loop review to check the chain of thought and solutions of the answers, then identidy the specific patterns into several categories:
1. Problem Restatement
2. Formula Selection & Symbolic Derivation
3. Step-by-Step Numeric Substitution
4. Multi-Path or Case Enumeration
5. Forward vs. Backward Reasoning
6. Self-Check & Validation
Then construct specific prompts for LLMs to identify the pattern of each answer.

## B TOKEN LEVEL ANALYSIS

We also perform token level analysis in the experiment. To quantify the internal certainty and decisiveness of reasoning models during CoT generation, we propose two token-level metrics: *average token confidence* and *average token gap*. These statistics offer fine-grained insight into the reliability of the model's reasoning process.

### B.1 AVERAGE TOKEN CONFIDENCE

Let the model generate a reasoning chain of $N$ tokens. The token confidence is defined by:

$$\ell_i = \log P(t_i \mid t_{<i}), \quad p_i = \exp(\ell_i)$$

$$\text{avg\_confidence} = \frac{1}{N}\sum_{i=1}^{N} p_i \quad (\times 100\%).$$

A higher average confidence reflects the model's self-assessed certainty in generating each step of its reasoning chain.

### B.2 AVERAGE TOKEN GAP

To assess decisiveness at each token step, we define the token gap as the difference between the top-1 and top-2 token probabilities:

$$g_i = \ell_i^{(1)} - \ell_i^{(2)}, \quad \text{avg\_token\_gap} = \frac{1}{N}\sum_{i=1}^{N} g_i$$

## C LLM USAGE

The authors would like to acknowledge the use of OpenAI's GPT-4.5 OpenAI (2023) for grammar polishing and language enhancement in this paper. The AI tool was used solely for improving the clarity and readability of the text, while all technical content, ideas, and conclusions remain the authors' own. We appreciate the advancements in AI-assisted writing tools that help researchers communicate their work more effectively.

## Prompt Template for Reasoning Pattern Analysis

Review the problem statement, the reference solution, and the model's chain-of-thought. Identify which one of the following high-level reasoning patterns the model employs, then output only the category number or name:

### *Problem Restatement & Known-Quantity Definition*

– The model starts by paraphrasing the question and listing all given variables with their symbols.

### *Formula Selection & Symbolic Derivation*

– The model names the governing law or equation, performs any algebraic rearrangements symbolically, then substitutes numbers.

### *Step-by-Step Numeric Substitution*

– The model breaks down each formula into small steps, plugs in values, computes intermediate results, and carries them forward.

### *Multi-Path or Case Enumeration*

– The model either runs two or more equivalent solution methods in parallel or enumerates multiple sign/geometric cases, then picks the valid result.

### *Forward vs. Backward Reasoning*

– *Forward*: from known data to answer step by step.
– *Backward*: start with the final condition/equation, then solve backward for the unknown.

### *Self-Check & Validation*

– After key steps, the model pauses to sanity-check units or compare parallel-path results before proceeding.

Figure 5: Prompt for analyzing reasoning patterns

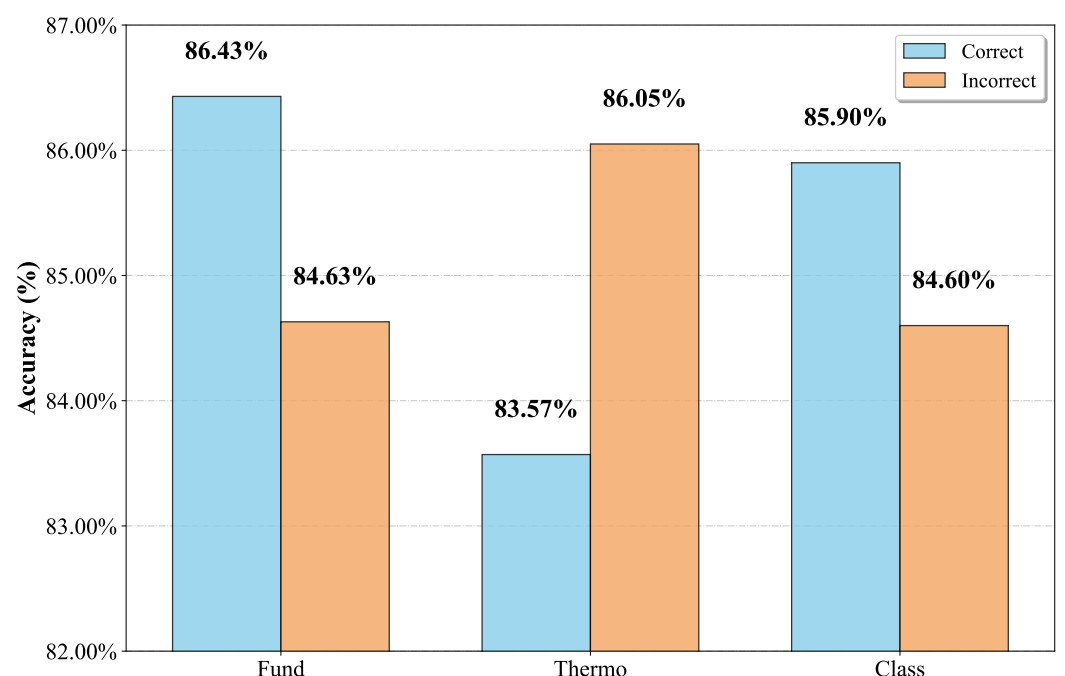

Figure 6: Average token confidence for correct vs. incorrect answers.

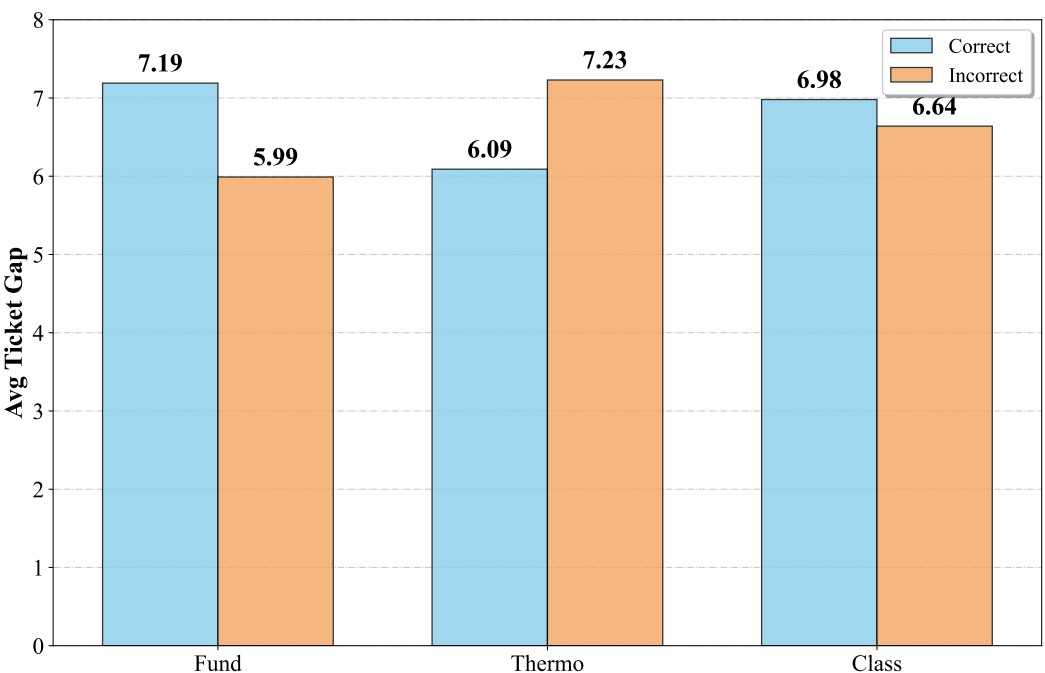

Figure 7: Average token gap for correct vs. incorrect answers.

