# OpenReview forum: "Symbolic or Numerical? Understanding Physics Problem Solving in Reasoning LLMs"
_ICLR.cc/2026/Conference — ICLR 2026 Conference Withdrawn Submission_

### Official Review · Reviewer_Zjhd · 2025-10-16

**Soundness:** 1
**Presentation:** 2
**Contribution:** 1
**Rating:** 0
**Confidence:** 5

**Summary:**

This paper conducts some experiments on a small subset of SciBench using a few Large Language Models (LLMs).
The main findings are:
1. The reasoning models are better than fast-thinking LLMs on physics reasoning problems.
2. Few-shot learning could also help Large Reasoning Models (LRMs)

**Strengths:**

I read this paper twice, and can not find any strengths.

**Weaknesses:**

1. Many Conceptual Errors:
+ I do not know why the authors call the reasoning models "instruction-tuned reasoning models." This is very unprofessional. They have many names, but the authors consistently write the wrong name throughout this paper.
+ Many conceptual errors in L145-L147. I can not get what the authors want to express. This sentence is clearly written by an LLM, possibly a very weak LLM.
+ In L43-44, "These recent models are specifically optimized through extensive instruction tuning and reinforcement learning from human feedback". They are not optimized through RLHF.

2. The authors are unfamiliar with related works.
+ All the references are on/before 2024 and mainly in prompting methods. I can not imagine a paper submitted to ICLR2026 has such old related works.
+ Many important related works in the field of physics problem-solving are not covered, to name a few: UGPhysics, DeepPHY.

3. Experiments are not comprehensive, and evaluation settings are wrong.
+ In L156-161, the authors use a 1e-30 temperature and sampling only once to test reasoning models. This is the wrong setting, and all experimental results are not reliable.
+ Experiments only cover DeepSeek Series reasoning models; more models should be evaluated.
+ It is confusing in L135-L136: "Additionally, we filter out problems with detailed solutions to ensure they can be used as few-shot prompts." Why and how?

4. Limited Contribution:
+ All key findings listed in this paper are common sense and have already been discussed in many tech reports.
+ No new method or new dataset is proposed, just evaluating existing models on existing datasets.

**Questions:**

See Weaknesses.

---

### Official Review · Reviewer_XvAJ · 2025-10-26

**Soundness:** 2
**Presentation:** 2
**Contribution:** 2
**Rating:** 2
**Confidence:** 4

**Summary:**

This paper empirically studies physics problem solving in instruction-tuned reasoning LLMs (DeepSeek-R1 and its distilled variants) on SciBench subsets, comparing zero-shot vs. few-shot chain-of-thought (CoT) prompting and contrasting them with chat-oriented models. The central behavioral finding is a divergence in solution styles—symbolic derivation (algebra first, numbers later) versus step-by-step numerical substitution—with the former associated with higher accuracy. The authors further analyze error categories (via SciBench’s taxonomy) and introduce token-level metrics (average token confidence and token gap), and they discuss CoT verbosity and computational cost.

**Strengths:**

1. Novel empirical observation: the study surfaces a clear behavioral phenomenon—reasoning-oriented models tend to adopt symbolic derivation in physics problem solving, which may be directly linked to better performance; few-shot CoT continues to provide gains.

2. Clear research question and reasonably described setup: the work explicitly focuses on “how LLMs solve physics problems,” differentiating zero-shot vs. few-shot CoT and paying attention to intermediate reasoning chains.

3. Goes beyond accuracy: the discussion of readability and reasoning style provides finer-grained insight than reporting correctness alone.

**Weaknesses:**

1. Limited methodological innovation and over-reliance on prompting: the work primarily probes models via prompt variants (zero-/few-shot CoT) and descriptive frequency analyses, without introducing new algorithms, training procedures, tools, or a formal methodology. The conclusions are largely correlational and risk conflating correlation with causation.

2. Fairness of comparisons is compromised: the evaluated subset is filtered (Section 3.2), yet baseline numbers are taken from Chen et al. (2023) under different settings/prompts and possibly different item pools (Table 2), undermining external validity.

3. Small evaluation size and low statistical power: the dataset comprises only a few hundred problems, with no confidence intervals, significance tests, multi-seed runs, or sensitivity analyses (e.g., temperature), making reported improvements vulnerable to noise and prompt-induced variance.

4. Outdated/incomplete baselines: the study includes few recent models and omits tool-augmented or programmatic baselines (e.g., equation writing + symbolic solvers, calculator-backed numeric pipelines), limiting conclusions about when symbolic derivation is preferable to numeric computation with tools.

5. Narrow scope of impact: the paper does not convincingly articulate how the finding informs model design, training strategies, or deployment (e.g., educational settings), limiting its practical significance beyond the observed phenomenon.

**Questions:**

Refer to the weaknesses.

---

### Official Review · Reviewer_EkFe · 2025-10-29

**Soundness:** 2
**Presentation:** 2
**Contribution:** 2
**Rating:** 2
**Confidence:** 4

**Summary:**

This paper, “Symbolic or Numerical? Understanding Physics Problem Solving in Reasoning LLMs”, explores how reasoning-tuned large language models like Deepseek-R1 and its distilled variants handle physics problems from the SciBench benchmark. The authors evaluate performance across three domains: fundamental physics, thermodynamics, and classical dynamics, under both zero-shot and few-shot chain-of-thought (CoT) prompting. They compare the results with standard models such as GPT-4 and Claude2, and claim that reasoning-oriented models achieve state-of-the-art accuracy while showing distinctive symbolic reasoning behavior. The paper also analyzes error types, reasoning patterns, and introduces some token-level confidence metrics.

**Strengths:**

The paper is clearly written and the experimental setup is well explained. The authors describe datasets, prompting methods, and evaluation criteria in detail, which makes the work reproducible. The focus on symbolic versus numerical reasoning is conceptually interesting and could inspire more interpretable studies of model reasoning behavior. The inclusion of both large and distilled models also gives a decent view of how scaling and compression affect reasoning. Visuals and tables are easy to read, and overall presentation quality is good.

**Weaknesses:**

- Limited novelty. The work mostly reuses existing datasets and prompting methods without introducing a clear methodological contribution. The idea of comparing symbolic and numerical reasoning is interesting but remains descriptive. There is no formal way to define or quantify what counts as “symbolic reasoning,” which weakens the main claim.
- Lack of deeper insight. The analysis of results is mainly observational. The authors describe trends but do not provide clear explanations or theoretical reasoning for why symbolic reasoning leads to higher accuracy. There is also no statistical link between reasoning type and performance.
- Experimental credibility. The performance gaps with prior baselines appear quite large, and it is unclear if all models were evaluated under identical setups. Some domains, especially thermodynamics, still show low accuracies, but the paper offers little analysis beyond suggesting that such problems are “more abstract.”
- Weak positioning. The related work section reads like a literature list rather than a motivation. The supposed contribution—understanding symbolic reasoning—is not clearly contrasted with existing research on reasoning interpretability or self-verification.
- Method limitations. The experiments use an extremely low temperature (1e-30), which may restrict reasoning diversity. The token-level confidence metrics in the appendix are mentioned but not properly interpreted. These parts feel incomplete and add little to the main conclusions.
- The study is well organized and clearly presented, but the contribution is mostly descriptive and lacks strong novelty or analysis. The findings about symbolic reasoning are interesting, yet the evidence remains anecdotal. The work feels more like a benchmark replication with commentary than a research paper offering new insights or methods.

**Questions:**

- Some models in Table 2 show very low accuracies, even for GPT-4 and Claude2. Were these re-evaluated or taken from prior reports? Could differences in parsing tolerance or formatting explain such large drops?
- How exactly is “symbolic reasoning” detected? Was it annotated by humans, or based on heuristics? Could the difference be just a matter of verbosity rather than reasoning depth?
- The paper concludes that symbolic reasoning improves performance, but no statistical evidence is given. Can you provide correlation or ablation analyses to support this claim?
- The reported token counts for Deepseek-R1 (often 10k+) are extremely high. How practical is this reasoning behavior in real-world use?
- How do you ensure fairness when comparing Deepseek-R1 with GPT-4 Turbo, given that model architectures, inference settings, and token budgets differ?

---

### Official Review · Reviewer_GwoV · 2025-10-30

**Soundness:** 2
**Presentation:** 2
**Contribution:** 2
**Rating:** 2
**Confidence:** 5

**Summary:**

This paper investigates whether modern large reasoning models, such as DEEPSEEK-R1 and its distilled variants, can effectively solve physics reasoning tasks without extensive prompt engineering or external tools. The authors benchmark these models on three datasets from SciBench (covering classical dynamics, thermodynamics, and fundamental physics) and compare their performance against more general chat-oriented LLMs.

The authors specifically explore - (1) Whether reasoning models can perform well intrinsically without heavy prompting; (2)  Whether few-shot prompt design still provides measurable benefits; (3) What underlying reasoning mechanisms distinguish reasoning models from standard chat models.

**Strengths:**

1.  Most existing studies on reasoning LLMs concentrate on mathematics, logic puzzles, or code synthesis. This paper’s focus on physics, thereby, broadening the empirical scope of reasoning model evaluation.
2.  The evaluation pipeline (zero-shot vs. few-shot CoT) is well described, using publicly available datasets (SciBench). The inclusion of multiple Deepseek variants and comparison with baselines like GPT-4-Turbo adds breadth.
3.  The paper includes appendices with prompt templates, parameter settings, and reproducibility statements, which are helpful for replication.
4.  The contrast between symbolic derivation and numeric substitution reasoning styles is somewhat insightful and well illustrated.

**Weaknesses:**

1.  This is primarily an evaluation work and lacks novelty. The authors evaluate pre-existing reasoning models (Deepseek-R1 and distill variants) on a known benchmark (SciBench).
2.  The “symbolic vs. numeric” observation, while intuitive, is anecdotal and not systematically analyzed or quantified.
3.  The evaluation is restricted to SciBench and unimodal, text-only physics questions, excluding diagrams, visual reasoning, or multimodal tasks that are central to real-world physics understanding.
4.  It is unclear whether Deepseek-R1’s pretraining might overlap with SciBench content leading to potential contamination.

Overall, this submission adds minimal incremental value and lacks novelty and significant contribution required to extend the current state-of-the-art.

**Questions:**

1.  Citations use inconsistent style
2.  In the accuracy computation, you allow for a 5% tolerance - why is that?
3.  "Qwen architecture's superior symbolic processing capabilities" - this is unclear.

---

### Note · Authors · 2025-12-01

I have read and agree with the venue's withdrawal policy on behalf of myself and my co-authors.